

# Molecular identification of *Trichobilharzia* species in recreational waters in North-Eastern Poland

Joanna Korycińska[1], Jana Bulantová[2], Petr Horák[2] and Ewa Dzika[1]

[1] Department of Medical Biology, School of Public Health, Collegium Medicum, University of Warmia and Mazury in Olsztyn, Olsztyn, Poland
[2] Department of Parasitology, Faculty of Science, Charles University, Prague, Czech Republic

## ABSTRACT

**Background**. In Europe, avian schistosomes of the genus *Trichobilharzia* are the most common etiological agents involved in human cercarial dermatitis (swimmer's itch). Manifested by a skin rash, the condition is caused by an allergic reaction to cercariae of nonhuman schistosomes. Humans are an accidental host in this parasite's life cycle, while water snails are the intermediate, and waterfowl are the final hosts. The study aimed to conduct a molecular and phylogenetic analysis of *Trichobilharzia* species occurring in recreational waters in North-Eastern Poland.

**Methodology**. The study area covered three water bodies (Lake Skanda, Lake Ukiel, and Lake Tyrsko) over the summer of 2021. In total, 747 pulmonate freshwater snails (*Radix* spp., *Lymnaea stagnalis*) were collected. Each snail was subjected to 1–2 h of light stimulation to induce cercarial expulsion. The phylogenetic analyses of furcocercariae were based on the partial sequence of the ITS region (ITS1, 5.8S rDNA, ITS2 and 28SrDNA). For *Radix* spp. phylogenetic analyses were based on the ITS-2 region.

**Results**. The prevalence of the *Trichobilharzia* species infection in snails was 0.5%. Two out of 478 (0.4%) *L. stagnalis* were found to be infected with *Trichobilharzia szidati*. Moreover, two out of 269 (0.7%) snails of the genus *Radix* were positive for schistosome cercariae. Both snails were identified as *Radix auricularia*. One of them was infected with *Trichobilharzia franki* and the other with *Trichobilharzia* sp.

**Conclusions**. Molecular identification of avian schistosome species, both at the intermediate and definitive hosts level, constitutes an important source of information on a potential threat and prognosis of local swimmer's itch occurrence, and helps to determine species diversity in a particular area.

# INTRODUCTION

Avian schistosomes of the genus *Trichobilharzia* are widely distributed parasites; the life cycle involves two hosts, the first being freshwater snails and the definitive host being waterfowl. Humans become accidental hosts when, following exposure to waters with the larval stage (cercariae), the parasites penetrate the skin (*Horák, Kolářová & Adema, 2002*). Cercarial dermatitis (CD), or swimmer's itch, occurs in many parts of the world, including the United States, China, Chile, Iran and Canada (*Brant & Loker, 2009*; *Wang et*

Corresponding author
Joanna Korycińska,
joanna.korycinska@uwm.edu.pl

*al., 2009*; *Valdovinos & Balboa, 2008*; *Farahnak & Essalat, 2003*; *Gordy, Cobb & Hanington, 2018*). In Europe, these cases have been reported in several countries, including Germany, Italy, Belgium, France, Switzerland, Austria, Norway, Iceland, the Netherlands, the United Kingdom (*Selbach, Soldánová & Sures, 2016*; *De Liberato et al., 2019*; *Caron et al., 2017*; *Caumes et al., 2003*; *Chamot, Toscani & Rougemont, 1998*; *Hörweg, Sattmann & Auer, 2006*; *Soleng & Mehl, 2010*; *Skírnisson & Kolářová, 2008*; *Schets et al., 2008*; *Kerr et al., 2024*), and also in Poland (*Marszewska et al., 2016*; *Korycińska et al., 2021*).

Cercarial dermatitis should be considered when diagnosing if a skin reaction occurs soon after exposure to recreational water. Initially, the dominating symptoms are maculo-papulo-vesicular eruptions accompanied by intense itching with typical over time progress (*Tracz et al., 2019*; *Macháček et al., 2018*). In some cases, generalized symptoms, such as nausea, diarrhea or fever may also occur. In extreme cases, respiratory distress and anaphylactic shock may be seen (*Bayssade-Dufour, Martins & Vuong, 2001*). Primary infections may be asymptomatic or limited in their manifestations to mild skin reactions with macules or maculopapules. Repeated reinfections result in sensitization phenomena leading to hypersensitivity reactions (*Kolářová et al., 2012*). As humans remain accidental hosts for avian schistosomes, the larval stage does not transform into the adult stage of the parasite. The research conducted to date has neither confirmed nor excluded further migration of the larval stage beyond the skin barrier in humans (*Macháček et al., 2018*; *Horák & Kolářová, 2001*). However, as studies involving mammalian models have demonstrated, cercariae of avian schistosomes managed to break the skin barrier and reach some organs, including the lungs, liver, heart, kidneys and spinal cord, and transform into schistosomula (*Horák & Kolářová, 2001*; *Lichtenbergová & Horák, 2012*). In the case of *T. regenti,* a high affinity to the host's central nervous system (CNS) can be seen. Experimental infections of mice have shown that, following penetration of the skin by cercariae of *T. regenti*, schistosomula can evade the attack by immune cells in the skin of mammalian hosts and reach the CNS, where immature forms of parasites die after several days (*Hrádková & Horák, 2002*; *Kouřilová, Syruček & Kolářová, 2004*). In these cases, both in definitive hosts (birds) and in mammalian hosts, some changes in tissues of the CNS were observed (*Hrádková & Horák, 2002*; *Kouřilová, Syruček & Kolářová, 2004*)). Other reported abnormalities included leg paralysis, balance and orientation disorders (*Lichtenbergová & Horák, 2012*; *Horák et al., 1999*; *Kolářová, Horák & Čada, 2001*).

*Trichobilharzia* species are responsible for most cases of cercarial dermatitis recorded in Europe (*Macháček et al., 2018*). About 40 species have been identified worldwide within the genus (*Horák, Kolářová & Adema, 2002*). Of these, six species have so far been reported in Europe, including: *T. szidati* (*Neuhaus, 1952*), *T. franki* (*Müller & Kimmig, 1994*), *T. regenti* (*Horák, Kolářová & Dvořák, 1998*), *T. salmanticencis* (*Simon-Martin & Simon-Vincente, 1999*), *T. anseri* (*Jouet et al., 2015*) and *T. mergi* (*Kolářová et al., 2013*). However, as research has progressed, another species, *T. physellae*, has been reported in Austria (*Helmer et al., 2021*).

Intermediate hosts comprise snails representing two main families of Lymnaeidae and Physidae. In Europe, *Trichobilharzia* spp. have, among others, been reported in *Lymnaea stagnalis* (Linnaeus, 1758), *Stagnicola palustris* (Müller, 1774), *Radix auricularia* (Linnaeus,

1758), *R. lagotis* (Schrank, 1803), *R. ampla* (Hartmann, 1821), *R. peregra* (Müller, 1774) (synonymous with *R. ovata* (Draparnaud, 1805), *R. balthica* (Linnaeus, 1758) and *R. labiata* (Rossmässler, 1835)) as well as *Physella acuta* (Draparnaud, 1805) (*Horák, Kolářová & Adema, 2002*; *Helmer et al., 2023*; *Bargues et al., 2001*). Definitive hosts, in turn, are some waterfowl species. For instance, in European water bodies, *T. regenti* has been detected in *Anas platyrhynchos* (Linnaeus, 1758), *Aythya fuligula* (Linnaeus, 1758), *Mergus merganser* (Linnaeus, 1758), and *A. clypeata* (Linnaeus, 1758), *T. szidati* in *A. crecca* (Linnaeus, 1758), *A. platyrhynchos*, and *A. clypeata*, while *T. franki* has been identified in *A. platyrhynchos*, *A. crecca*, *A. fuligula*, and *Cygnus olor* (Gmelin, 1789) (*Jouet et al., 2010a*; *Jouet et al., 2009*; *Jouet et al., 2010b*; *Skírnisson, Aldhoun & Kolářová, 2009*).

In temperate climate zones, most CD cases are reported in warm summer months, mainly associated with an increase in recreational water activities. Some studies indicate there is a greater risk of infection in individuals aged below 15 (spending more time in shallow, warm water, where snails and parasites are found), while others exclude age risk factors (*Chamot, Toscani & Rougemont, 1998*; *Hörweg, Sattmann & Auer, 2006*; *Verbrugge et al., 2004*; *Lindblade, 1998*; *Soldánová et al., 2013*). Other important swimmer's itch risk factors are bathing activity and the time spent in the water (*Chamot, Toscani & Rougemont, 1998*; *Verbrugge et al., 2004*; Lindblade).

In Poland, there is no continuous monitoring and recording of CD cases over a particular area. Therefore, information concerning the occurrence of CD comes from individual reports, physicians and the State Sanitary Inspectorate. In the summer season, the quality of water at bathing sites is tested by the State Sanitary Inspectorate. The monitoring is conducted to detect possible microbiological threats, such as *Escherichia coli*, *Enterococcus* or cyanobacteria, macroalgae and phytoplankton. The research on CD exceeds the range of the tests conducted by the Sanitary Inspectorate. A dermatitis outbreak in a particular area requires a series of steps to be taken. First, the snails should be collected and screened for cercariae, which may be responsible for skin lesions. Alternatively, the cercariae or their e-DNA can be collected and concentrated directly from water samples (*Helmer et al., 2023*; *Rudko et al., 2019*). Other detection methods include the analysis of bird droppings for parasite eggs and autopsies of birds (*Horák et al., 2015*). If avian schistosomes are identified in birds, there is a need to check if suitable intermediate hosts for cercarial development are present in a particular water body.

It must be emphasized that identifying species based on their morphological and anatomical features is insufficient, particularly for the genus *Trichobilharzia* (*Horák et al., 2012*). Therefore, identification requires molecular analyses. The analyses conducted so far have been based on three gene regions, including 28S, the internal transcribed spacer (ITS) and the mitochondrial cox1. The investigations have confirmed that the nuclear ITS regions proved effective in delineating among *Trichobilharzia* species that are responsible for outbreaks of cercarial dermatitis (*Davis, Blair & Brant, 2021*; *Japa et al., 2021*; *Lashaki et al., 2023*; *Ashrafi et al., 2021*).

This study aimed to identify schistosome species of the genus *Trichobilharzia* in the intermediate hosts of the Lymnaeidae family in three recreational water bodies in North-Eastern Poland.

## MATERIALS AND METHODS

### Cercariae and snail examination

The study involved three recreational water bodies within administrative boundaries of the City of Olsztyn: Lake Ukiel covering 412 ha (53°47′38.4″N 20°25′55.4″E), Lake Skanda covering 51.5 ha (53°45′40.3″N 20°31′34.9″E) and Lake Tyrsko of 18.6 ha (53°48′21.1″N 20°25′06.9″E) (Fig. 1). The lakes have an important recreational function during the summer season. The nearshore zones of the lakes are lined with *Phragmites australis*, *Acorus calamus* and *Schoenoplectus lacustris*, while submerged vegetation is mainly represented by *Ceratophyllum demersum*, *Elodea canadensis* and *Potamogeton natans*. Moreover, waterfowl were observed at bathing sites during sampling. The most often-seen species include *A. platyrhynchos*, *C. olor*, *Fulica atra* and *Podiceps* sp.

For each lake, snails were obtained from one selected site, providing the best beach recreation opportunities and access to water. In the summer of 2021, from June to August, 747 pulmonate freshwater snails were collected (*L. stagnalis* and *Radix* spp.).

The snails were collected by hand or using a plastic sieve during morning hours. They were subsequently carried to a laboratory after being placed in plastic containers filled with water. Preliminary identification of snail species was conducted using a key (*Piechocki & Wawrzyniak-Wydrowska, 2016*). Each snail was placed in a glass beaker with dechlorinated tap water and subjected to 1–2 h of light stimulation to induce cercarial expulsion. Cercariae were identified according to morphological criteria using a light microscope (*Combes et al., 1980*). Furcocercariae with pigmented eye spots were collected into 1.5 ml tubes with 95% molecular grade ethanol and frozen (−20 °C) until DNA extraction was performed. In the case of *Radix* spp. snails, for DNA extraction, a small soft piece of the posterior part of the foot was separated. The samples were stored at −70 °C for future analysis.

### DNA extraction

Before cercarial DNA extraction, the material was centrifuged, ethanol was removed and the samples were left at room temperature until the alcohol evaporated completely. This step was followed by adding 300 μl nuclease-free water, 300 μl lysis buffer (A&A Biotechnology, Gdynia, Poland) and 20 μl proteinase K (A&A Biotechnology, Gdynia, Poland). It was then incubated with shaking at 400 rpm in Thermomixer Comfort (Eppendorf) for 2 h at 50 °C. Total DNA was extracted according to the manufacturer's tissue protocol (Sherlock AX, A&A Biotechnology, Gdynia, Poland). DNA was eluted in 50 μl of the elution buffer TE provided and stored at −70 °C.

Parts of the snail foot were defrosted, placed on a Petri dish and washed 3 times with PBS buffer. The next stage was the extraction of DNA from infected snails of the genus *Radix*. A tissue sample (no more than 25 mg) was cut into small pieces using a scalpel and placed in a 1.5 ml microcentrifuge tube. Following the addition of 180 μl of lysis buffer and 20 μl proteinase K (Qiagen, Hilden, Germany), the sample was incubated at 56 °C until completely lysed (24 h) in Thermomixer Comfort (Eppendorf) with shaking at 500rmp. DNA extraction was conducted using the Qiamp DNA Mini Kit (Qiagen, Hilden, Germany) according to the manufacturer's protocol. DNA was then eluted in 50 μl of the elution buffer AE provided and stored at −70 °C.

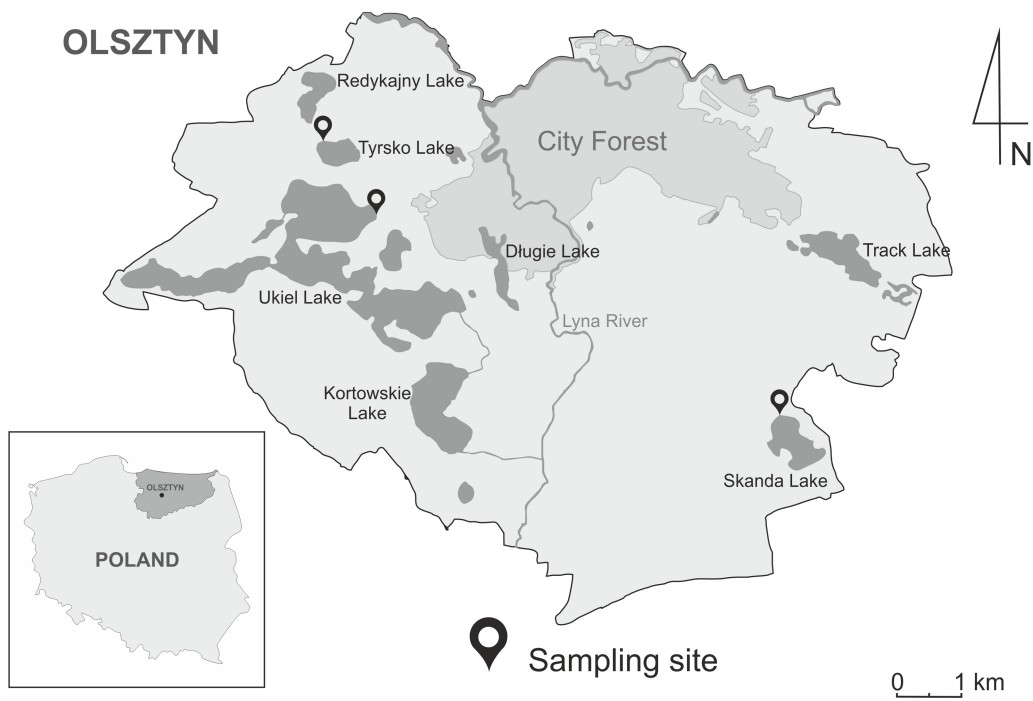

**Figure 1** **Geographic map of the Olsztyn, North-Eastern Poland, showing three localities of sampling sites of snails in the years 2020–2021.** The map was designed in CorelDRAWX5 based on Google Maps (https://www.google.pl/maps).

## Molecular identification and sequencing

The ITS gene region of cercariae was amplified using primers its5Trem (5′-GGAAGTAAAAGTCGTAACAAGG-3′) complementary to the conserved region at the 3′end of the 18SrRNA gene and its4Trem (5′-TCCTCCGCTTATTGATATGC-3′) complementary to the conserved region at the 5′end of the 28S rRNA gene (*Dvořák et al., 2002*). The identification of *Radix* spp. snails positive for *Trichobilharzia* spp. was performed using primers LT1 (5′-TCGTCTGTGTGAGGGTCG-3′) and ITS2-RIXO (5′-TTCTATGCTTAAATTCAG GGG-3′) (*Bargues et al., 2001*). Polymerase chain reaction for cercariae was performed in Mastercycler Nexus (Eppendorf, Hamburg, Germany) in a total volume of 25 µl. The PCR mixture contained 15–100 ng DNA, 0.2 µM of each primer, 2.5 µl of 10X Ex Taq Buffer ($Mg^{2+}$ free), 2.0 µl $MgCl_2$ (25 mM), 2.0 µl dNTP Mixture (2.5 mM each) and 0.13 µl TaKaRa Ex Taq (5 units/µl).

The conditions for the reaction were as follows: initial denaturation at 95 °C for 5 min, 35 cycles of denaturation at 95 °C for 60 s, annealing at 50 °C for 45 s, extension at 72 °C for 120 s, followed by final extension at 72 °C for 10 min (*Dvořák et al., 2002*). For snails, the 25 µl PCR mixture contained 12.5 µl of Go Taq Green Master MIX (Promega, Medison, USA), 9.5 µl nuclease-free water, 1 µl of each primer (10 µM) and 1 µl of template DNA. The thermal conditions of the PCRs were as follows: 2 min at 94 °C, by 30 cycles of 30 s at 94 °C, 30 s at 50 °C, 1 min at 72 °C. Finally, an extension step of 7 min was performed at 72 °C (*Bargues et al., 2001*).
The obtained PCR products were visualized using electrophoresis in 1% agarose gel with the added Midori Green Advance (Nippon Genetics, Japan). Purified with Clean-Up (A&A Biotechnolgy, Gdynia, Poland), the products underwent sequencing in both directions by Macrogen Humanizing Genomics Europe (Amsterdam, The Netherlands). Consensus sequences were aligned using BioEdit Sequence Alignment Editor *v.* 7.1.10 (*Hall, 1999*) and analyzed using BLAST (http://www.ncbi.nlm.nih.gov/BLAST). Four sequences for cercariae (OP681463, OP648160, OP681454, OP681462) and 2 sequences for snails (OQ645742, OQ645741) were deposited in GenBank.

### Phylogenetic analyses

The phylogenetic analyses for furcocercariae were based on the partial sequence of the ITS region (ITS1, 5.8S rDNA, ITS2, and 28SrDNA). The tree was constructed using two isolates from Lake Ukiel (UL1, UL2), one isolate from Lake Skanda (SL1), one from Lake Tyrsko (TL1) and 19 sequences of *Trichobilharzia* species from the GenBank database  (Table 1). *Schistosoma edwardiense*, *S. hippopotami*, *Schistosomatidae* sp.*, Gigantobilharzia huronensis* and *Dendritobilharzia pulverulenta* sequences were used as an outgroup to root the trees. For *Radix* spp. snails, phylogenetic analyses of the ITS-2 region were performed using the 14 sequences downloaded from GenBank and two obtained in this study (isolates ULS1, ULS2)  (Table 2). Sequences of *L. stagnalis* served as an outgroup. Sequences were aligned using Muscle (*Edgar, 2004*) in Mega X (*Kumar et al., 2018*) and trimmed at the ends using BioEdit *v.* 7.1.10. The trees were constructed using Mega X software (*Kumar et al., 2018*).

The maximum likelihood (ML) method was applied using the Tamura 3-parameter (T92+G) for snails and the Kimura 2-parameter (K2+G) for furcocercariae. Accordingly, there were 290 and 973 positions in the final dataset. In both analyses, the internal node support was assessed by 1,000 bootstrap sampling. The scale bar indicates the equivalence of distance between sequences.

## RESULTS

In total, 747 snails were collected in three water bodies, four of which (0.52%) revealed cercariae of the genus *Trichobilharzia* after illumination. Two out of 478 *L. stagnalis* (0.4%) and two out of 269 *Radix* spp. (0.7%) were found infected. There were 2 infected *R. auricularia* snails identified in Lake Ukiel (locality 1), while only one infected *L. stagnalis* was reported for Lake Skanda (locality 2) and Lake Tyrsko (locality 3) likewise (Table 3).

In molecular analyses of cercariae, the phylogenetic tree (Fig. 2) was based on the ITS region of ribosomal DNA. *Trichobilharzia* spp. isolates split into several groups. The first cluster contains *T. franki* and *T. regenti*. The second includes sequences of *T. szidati*, *T. stagnicolae* and one clade of undetermined sequences of *Trichobilharzia* sp.

The analysis showed that the four isolates identified in this study belonged to the genus *Trichobilharzia*. Two isolates, SL1 and TL1 (GenBank: OP648160; OP681454), showed nearly 100% similarity to *T. szidati* isolates from Czechia (AY713972, GU233735); Denmark (KP271014) and Poland (MT041670). It was determined that UL1 isolate (GenBank: OP681463) showed nearly 100% similarity to *T. franki* from Czechia (AF356845,

**Table 1  List of Gen Bank sequences of bird schistosomes used for phylogenetic analysis.**

| Isolate based on ITS region | Accessions numbers | Host | Stage | Country of isolation | Reference |
|---|---|---|---|---|---|
| *Trichobilharzia franki* | AF356845 | *Radix auriculari* | Cercaria | Czechia | *Dvořák et al. (2002)* |
| *Trichobilharzia franki* | MW482441 | *Ampullaceana balthica* | Cercaria | Denmark | *Al-Jubury et al. (2021)* |
| *Trichobilharzia franki* | MW538531 | *Ampullaceana balthica* | Cercaria | Denmark | *Al-Jubury et al. (2021)* |
| *Trichobilharzia franki* | MW482439 | *Ampullaceana balthica* | Cercaria | Denmark | *Al-Jubury et al. (2021)* |
| *Trichobilharzia franki* | AY713964 | *Radix auricularia* | Cercaria | Poland | Unpublished |
| *Trichobilharzia franki* | AY713969 | *Radix auricularia* | Cercaria | Czechia | Unpublished |
| *Trichobilharzia franki* | UL1 | *Radix auricularia* | Cercaria | Poland | Current study |
| *Trichobilharzia regenti* | EF094537 | *Anas platyrhynchos* | Egg | Poland | *Rudolfova et al. (2007)* |
| *Trichobilharzia regenti* | EF094534 | *Aythya fuligula* | Egg | Poland | *Rudolfova et al. (2007)* |
| *Trichobilharzia regenti* | KP271015 | *Radix peregra* | Cercaria | Denmark | *Christiansen et al. (2016)* |
| *Trichobilharzia regenti* | EF094535 | *Anas platyrhynchos* | Egg | Poland | *Rudolfova et al. (2007)* |
| *Trichobilharzia regenti* | AF263829 | *Radix peregra* | Cercaria | Czechia | *Dvořák et al. (2002)* |
| *Trichobilharzia szidati* | MW143565 | *Lymnaea stagnalis* | Cercaria | Austria | *Gaub et al. (2020)* |
| *Trichobilharzia szidati* | MW143565 | *Lymnaea stagnalis* | Cercaria | Austria | *Gaub et al. (2020)* |
| *Trichobilharzia szidati* | AY713972 | *Lymnaea stagnalis* | Cercaria | Czechia | *Rudolfová et al. (2005)* |
| *Trichobilharzia szidati* | SL1 | *Lymnaea stagnalis* | Cercaria | Poland | Current study |
| *Trichobilharzia szidati* | TL1 | *Lymnaea stagnalis* | Cercaria | Poland | Current study |
| *Trichobilharzia* sp. | FJ469793 | *Radix peregra* | Cercaria | Iceland | *Aldhoun et al. (2009)* |
| *Trichobilharzia* sp. | FJ469797 | *Mergus serrator* | Adult | Iceland | *Aldhoun et al. (2009)* |
| *Trichobilharzia* sp. | UL2 | *Radix auricularia* | Cercaria | Poland | Current study |
| *Trichobilharzia stagnicolae* | FJ174540 | *Stagnicola* sp. | Cercaria | USA | *Brant & Loker (2009)* |
| *Trichobilharzia stagnicolae* | FJ174541 | *Stagnicola* sp. | Cercaria | USA | *Brant & Loker (2009)* |
| *Trichobilharzia stagnicolae* | FJ174543 | *Stagnicola* sp. | Cercaria | USA | *Brant & Loker (2009)* |
| *Schistosomatidae* sp. | KC113097 | *Chilina dombeiana* | Cercaria | Argentina | *Flores, Brant & Loker (2015)* |
| *Gigantobilharzia huronensis* | EF071986 | *Agelaius phoeniceus* | Adult | USA | *Brant (2007)* |
| *Dendritobilharzia pulverulenta* | AY713962 | *Gallus gallus* | Adult | USA | *Rudolfová et al. (2005)* |
| *Schistosoma edwardiense* | AY197344 | *Biomphalaria sudanica* | Cercaria | Uganda | *Morgan et al. (2003)* |
| *Schistosoma hippopotami* | AY197343 | *Bulinus truncatus* | Cercaria | Uganda | *Morgan et al. (2003)* |

AY713969) and Poland (AY713964). In turn, UL2 isolate (GenBank: OP681462) was nearly 100% similar to *Trichobilharzia* sp. from Iceland (FJ469793, FJ469797, FJ469792) (Table 4).

The phylogenetic tree of snail species was based on the ITS-2 domain of rDNA of snail isolates (Fig. 3). Sequences of *L. stagnalis* served as outgroups. Snail isolates split into several clades: *R. auricularia + R. labiata + R. ampla,* and *R. lagotis + R. peregra.* Two infected snails (ULS1 and ULS2 isolates) belonged to the *R. auricularia* group. ULS1 isolate (GenBank: OQ645742) showed 100% similarity to *R. auricularia* from Germany (LR738438, LT623582) and Czechia (GU574285), whereas the ULS2 isolate (GenBank: OQ645741) showed 100% similarity to *R. auricularia* from Germany (LT623582, H6931933) and Iran (KT365872) (Table 4).

**Table 2** List of Gen Bank sequences of snails used for phylogenetic analysis.

| Isolate based on ITS-2 region | Accessions numbers | Country of isolation | Reference |
|---|---|---|---|
| *Radix peregra* | AJ319633 | France | *Bargues et al. (2001)* |
| *Radix peregra* | AJ319634 | France | *Bargues et al. (2001)* |
| Radix peregra | GU574212 | Iceland | *Huňová et al. (2012)* |
| *Radix peregra* | GU574222 | Czechia | *Huňová et al. (2012)* |
| *Radix ampla* | HE573072 | Germany | *Schniebs et al. (2013)* |
| *Radix lagotis* | HE573077 | Germany | *Schniebs et al. (2013)* |
| *Radix lagotis* | HE573075 | Germany | *Schniebs et al. (2013)* |
| *Radix labiata* | AJ319636 | Czechia | *Bargues et al. (2001)* |
| *Radix labiata* | AJ319637 | Germany | *Bargues et al. (2001)* |
| *Radix auricularia* | GU574285 | Czechia | *Huňová et al. (2012)* |
| *Radix auricularia* | HG931933 | Unpublished | Unpublished |
| *Radix auricularia* | KT365872 | Iran | Unpublished |
| *Radix auricularia* | LR738438 | Estonia | Unpublished |
| *Radix auricularia* | LS974218 | Bulgaria | *Schniebs et al. (2013)* |
| *Radix auricularia* | ULS2 | Poland | Current study |
| *Radix auricularia* | ULS1 | Poland | Current study |
| *Lymnaea stagnalis* | FR797836 | Germany | *Vinarski et al. (2011)* |
| *Lymnaea stagnalis* | FR797834 | Germany | *Vinarski et al. (2011)* |

**Table 3** The number of snails infected with *Trichobilharzia* species in the examined lakes.

| | | Sampling site for snails | | | |
|---|---|---|---|---|---|
| | | Locality 1 Lake Ukiel | Locality 2 Lake Skanda | Locality 3 Lake Tyrsko | Total |
| Number of examined snails | *L. stagnalis* | 94 | 334 | 50 | 478 |
| | *Radix* sp. | 205 | 29 | 35 | 269 |
| | Total | 299 | 363 | 85 | 747 |
| Snails infected with *Trichobilharzia* species | *L. stagnalis* | – | *Ts* (1) | *Ts* (1) | 2 (0.4%) |
| | *R. auricularia* | *Tf* (1); *T* sp. (1) | – | – | 2 (0.7%) |
| | Total | 2 (0.6%) | 1 (0.2%) | 1 (1.1%) | 4 (0.5%)[*] |

**Notes.**
  Tf, *Trichobilharzia franki*; Ts, *Trichobilharzia szidati*; Tsp, *Trichobilharzia* species.
  [*]total prevalence including all the snails collected from the three lakes.

# DISCUSSION

Most avian schistosomes reported in European water bodies represented either of the two genera, *Trichobilharzia* or *Bilharziella* (*Helmer et al., 2023*; *Lashaki et al., 2020*). Generally, the prevalence of infections caused by *Trichobilharzia* species in intermediate hosts is not high and does not exceed a few percent. For instance, the studies conducted in Denmark revealed a prevalence of 0.6%, in Austria 0.4%, in France 0.1%, and in Germany 0.1% (*Reier et al., 2020*; *Jouet et al., 2008*; *Ferté et al., 2004*; *Loy & Haas, 2001*). Nevertheless, regionally, the prevalence may be much higher, *e.g.*, in the Netherlands at 5 to 12% (*Schets, Lodder*

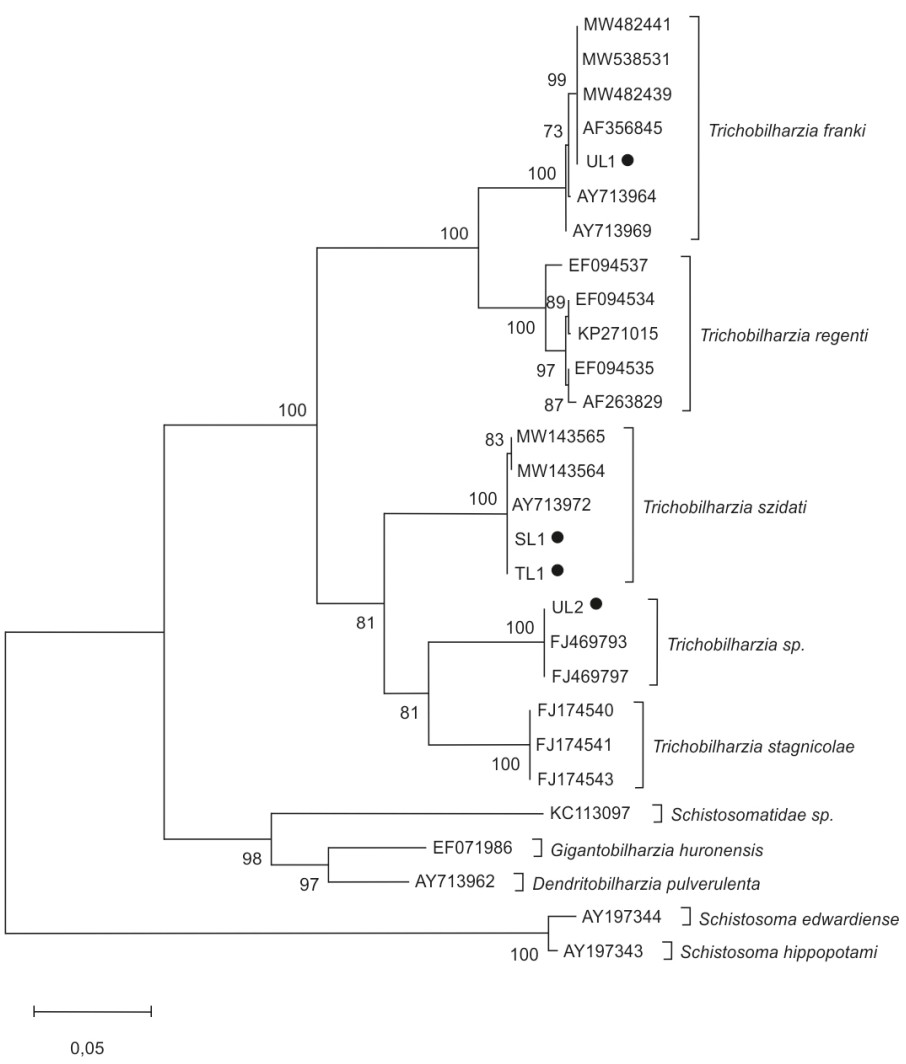

**Figure 2** **Phylogenetic tree based on the ITS rDNA of bird schistosomes.** The phylogenetic tree was constructed using the maximum likelihood method and K2+G substitution model. Sequences of *Schistosoma edwardiense, Schistosoma hippopotami, Schistosomatidae* sp.*, Gigantobilharzia huronensis, Dendritobilharzia pulverulenta* served as outgroups. The scale shows the number of nucleotide substitutions per site between DNA sequences. The node support is given in maximum likelihood bootstraps, maximum parsimony bootstraps and Bayesian posterior probability. The isolates obtained in this study are marked with a dot.

*& De Roda Husman, 2009*) or in Iceland at 24.5% (*Skírnisson, Aldhoun & Kolářová, 2009*). As far as definitive hosts are concerned, the prevalence is higher. The studies from France (*Jouet et al., 2008*) showed a 70% infection rate in *A. platyrhynchos*, while in *A. fuligula* and *A. farina*, 50% and 14%, respectively. In Iceland, the prevalence was 66.7–73.3% in *A. platyrhynchos*, 83.3% in *M. serrator* and 58.3% in *A. anser* (*Skírnisson, Aldhoun & Kolářová, 2009*).

Our study identified other busy recreational and bathing sites in North-Eastern Poland and reported the schistosome species found there to potentially cause human cercarial

**Table 4  Comparison of the obtained sequences with GenBank sequences.**

| Sequence ID/Isolate | Closest match ID | Origin | Percentage match | Query cover |
|---|---|---|---|---|
| | AF356845 | Czechia | 99.93% | 100% |
| UL1 | AY713969 | Czechia | 99.46% | 100% |
| | AY713964 | Poland | 99.46% | 100% |
| | AY713972 | Czechia | 99.60% | 100% |
| SL1 | KP271014 | Denmark | 99.52% | 100% |
| | GU233735 | Czechia | 99.52% | 100% |
| | MT041670 | Poland | 99.92% | 100% |
| TL1 | AY713972 | Czechia | 99.92% | 100% |
| | KP271014 | Denmark | 99.83% | 100% |
| | FJ469793 | Iceland | 99.93% | 100% |
| UL2 | FJ469797 | Iceland | 99.86% | 100% |
| | FJ469792 | Iceland | 99.86% | 100% |
| | LR738438 | Germany | 100% | 100% |
| ULS1 | LT623582 | Germany | 100% | 100% |
| | GU574285 | Czechia | 100% | 100% |
| | KT365872 | Iran | 100% | 100% |
| ULS2 | LT623582 | Germany | 100% | 100% |
| | HG931933 | Germany | 100% | 100% |

dermatitis. The infection rate of intermediate hosts with avian schistosomes was reported at 0.52%, corresponding to the above results. In previous studies conducted in Poland, the infection rate of schistosomes in intermediate hosts was reported at 1.8% −1.24% (*Marszewska et al., 2016*; *Marszewska et al., 2018*). In this study, the infection rates with *Trichobilharzia* species in *Radix* sp. and *L. stagnalis* were 0.7 and 0.4%, respectively. Of the 478 collected *L. stagnalis* snails, two (0.4%) were found to host *T. szidati*. Out of 269 *Radix* spp. snails, two *R. auricularia* (0.7%) were found to be infected with *T. franki* and *Trichobilharzia* sp. In other European countries, the infection rates in *L. stagnalis* with cercariae of *T. szidati* are similar. For instance, it is 2.96% in Germany, 0.5% in Denmark and 0.4% in France (*Selbach, Soldánová & Sures, 2016*; *Ferté et al., 2004*; *Christiansen et al., 2016*). In turn, the infection rate in *Radix* sp. in Denmark was 1.7% and in France 0.05% (*Ferté et al., 2004*; *Christiansen et al., 2016*).

According to the evidence presented in the literature, identification of cercariae of *Trichobilharzia* genus based on methods other than molecular is a tedious process posing difficulties to researchers (*Horák, Kolářová & Adema, 2002*). For instance, *Podhorský et al. (2009)* recommends that the morphological comparison of cercariae of *T. szidati, T. franki,* and *T. regenti* should be based on the distribution of sensory papillae and not on body measurements. Moreover, molecular analysis of the ITS region helped to systematize terminology for *T. szidati/T. ocellata* isolates reported in Europe (*Rudolfová et al., 2005*). Considering the above, it is of primary importance to conduct DNA analysis and compare the results with those presented in databases.

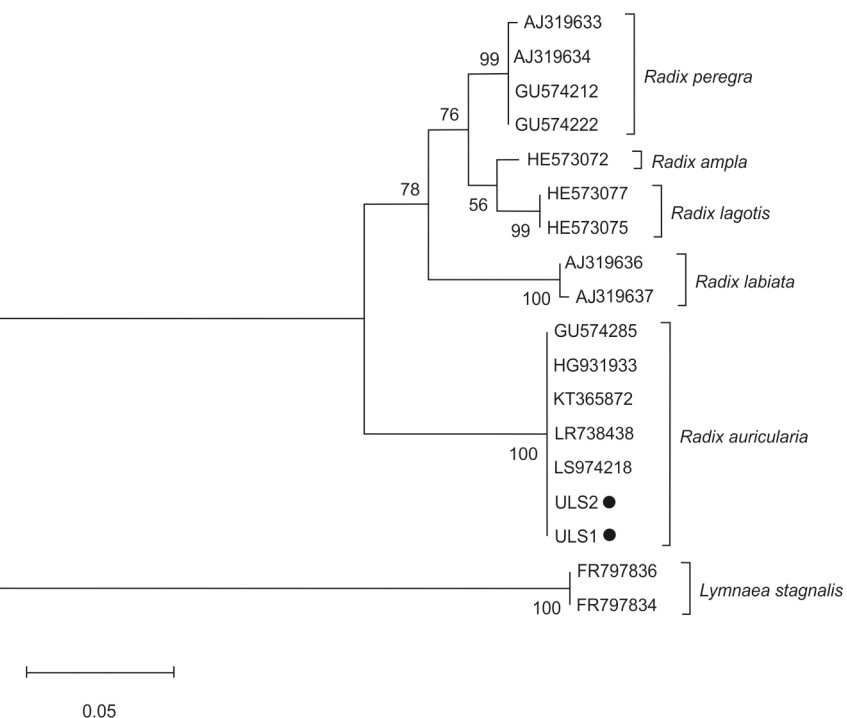

**Figure 3** **Phylogenetic tree based on the ITS-2 domain of rDNA of snails.** The phylogenetic tree was constructed using the maximum likelihood method and T92+G substitution model. Sequences of *L. stagnalis* served as outgroups. The scale shows the number of nucleotide substitutions per site between DNA sequences. The node support is given in maximum likelihood bootstraps, maximum parsimony bootstraps and Bayesian posterior probability. The isolates obtained in this study are marked with a dot.

The phylogenetic analysis shows that our study's *T. franki* and *T. szidati* sequences form a well-supported clade, separated from any other taxa, with almost 100% similarity to Czechia, Poland and Denmark sequences. *Lawton et al. (2014)* demonstrated a high population variation within and between *T. franki* populations. Likewise, in *T. szidati* isolates a high haplotype diversity was confirmed and mixed across their geographical origin (*Korsunenko et al., 2011*).

Research shows that the species representing *Trichobilharzia* are characterized by a high specificity for their intermediate hosts. It is usually one specific snail species or several closely related species (*Jouet et al., 2010b*; *Jouet et al., 2008*; *Kock, 2001*). The taxonomic distinction of species in the genus *Radix* should not exclusively be based on shell size and shape because they are phenotypically plastic in response to environmental conditions (*Pfenninger, Cordellier & Streit, 2006*). In our research, the phylogenetic analyses for *Radix* spp. snails were based on the ITS-2 region. In both cases, they were identified as *R. auricularia*.

In this study, *T. franki* was found in its typical host, *R. auricularia*. However, this particular species may also be seen in other intermediate hosts. In a study by *Aldhoun et al. (2009)* and *Jouet et al. (2010b)*, the host for *T. franki* was *R. peregra*.

Under laboratory conditions, it was demonstrated that miracidia of *T. franki* were able to infect 100% of *R. auricularia* and 9.8% of *R. ovata* specimens (probably corresponding to *R. peregra* sensu (*Bargues et al., 2001*; *Kock, 2001*). Moreover, *Jouet et al. (2010b)* demonstrated that *T. franki* from *R. auricularia* and *T. franki* from *R. peregra* belong to distinct clades. Molecular differences were observed within different domains, such as D2 and ITS of the ribosomal DNA, as well as cox1 of the mitochondrial DNA. Consequently, there is a rationale for reconsidering the membership of the species *T. franki* for the haplotypes isolated from *R. peregra*. What is also observed in Europe is the mixing of *T. franki* populations, with one of the haplotypes from the UK appearing in France, Switzerland and Czechia. This situation is probably connected with waterfowl migration (*Lawton et al., 2014*).

The main intermediate host for *T. szidati* remains *L. stagnalis*; its cercariae were also found in *S. palustris* (*Rudolfová et al., 2005*; *Kock, 2001*).

In the life cycle of *T. franki* and *T. szidati* the number of definitive hosts (waterfowl) seems to be more varied in comparison to intermediate hosts (*Horák, Kolářová & Adema, 2002*). In the localities studied, the observed species included, among others, *A. platyrhynchos*, *C. olor*, *F. atra* and *Podiceps* sp., which may become potential definitive hosts. Nevertheless, in order to confirm that, research should be extended and, additionally, postmortem parasitological examinations should be conducted in waterfowl. As our study and the research by other authors have shown, molecular identification of avian schistosome species in snails constitutes an important source of information concerning the local threat of a CD outbreak. To date, the species occurring in Poland that are at present recognized as potentially causing swimmer's itch include *T. szidati*, *T. franki* and *T. regenti*, with *T. szidati* being the most commonly found in water bodies (*Marszewska et al., 2016*; *Korycińska et al., 2021*; *Marszewska et al., 2018*; *Zbikowska, 2004*; *Zbikowska et al., 2006*). There have been a few confirmed cases of swimmer's itch in humans, including the Dzierzęcinka River -Water Valley (*Marszewska et al., 2016*) and Lake Pluszne in North-Eastern Poland (*Korycińska et al., 2021*). Moreover, in 2023 there was a report of CD cases when the condition followed bathing in Lake Drawskie (information obtained from materials published by the State Sanitary Inspectorate).

It is also important to note that progressing climate change directly affects intermediate and definitive hosts. A rise in water temperature accelerates both the development of freshwater snails and algae, which results in the increase in population of intermediate hosts and, therefore, a higher number of cercariae can be released. There are also fewer birds migrating to the south, which results in prolonged contact of the parasite with its definitive host and makes it more likely for the parasite to complete its life cycle (*Mas-Coma, Valero & Bargues, 2009*).

## CONCLUSIONS

This study identifies more bathing sites where *Trichobilharzia* species potentially causing swimmer's itch have been found. An important aspect is to investigate the genetic diversity of avian schistosomes over a particular area, involving both intermediate and definitive

hosts. Notably, there needs to be clear regulations regarding the prevention and monitoring of cercarial dermatitis. Actions aiming to reduce the threat are only taken in response to publicly reported cases and in locations of CD outbreaks. There is a rationale for the introduction of seasonal monitoring over the areas of increased recreational potential, particularly in places that have reported cercarial dermatitis. Also, consideration should be given to the idea of launching educational campaigns to spread awareness of the threat. This activity would allow suitable preventive measures to be taken and thus contribute to a lower incidence of cercarial dermatitis.

### Funding
The research of Petr Horák and Jana Bulantová is currently supported by the Czech Science Foundation (GA24-11031S), the European Regional Development Fund and Ministry of Education, Youth and Sports of the Czech Republic (CZ.02.1.01/0.0/0.0/16_019/ 0000759), and the Charles University institutional support (Cooperatio Biology, and UNCE24/SCI/011). The funders had no role in study design, data collection and analysis, decision to publish, or preparation of the manuscript.

### Grant Disclosures
The following grant information was disclosed by the authors:
Czech Science Foundation: GA24-11031S.
European Regional Development Fund and Ministry of Education, Youth and Sports of the Czech Republic: CZ.02.1.01/0.0/0.0/16_019/ 0000759.
Charles University: UNCE24/SCI/011.

### Competing Interests
The authors declare there are no competing interests.

### Author Contributions
- Joanna Korycińska conceived and designed the experiments, performed the experiments, analyzed the data, prepared figures and/or tables, authored or reviewed drafts of the article, approved the final draft, and approved the final draft.
- Jana Bulantová conceived and designed the experiments, analyzed the data, authored or reviewed drafts of the article, approved the final draft, and approved the final draft.
- Petr Horák conceived and designed the experiments, analyzed the data, authored or reviewed drafts of the article, approved the final draft, and approved the final draft.
- Ewa Dzika conceived and designed the experiments, analyzed the data, authored or reviewed drafts of the article, approved the final draft, and approved the final draft.

### DNA Deposition
The following information was supplied regarding the deposition of DNA sequences:
The sequences are available at GenBank: OP681463, OP648160, OP681454, OP681462, OP681742, OP681741.

## Data Availability

Raw data are available in Supplementary Files.

## Supplemental Information

Supplemental information for this article can be found online at http://dx.doi.org/10.7717/peerj.17598#supplemental-information.

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
