# Peer review of "Molecular identification of Trichobilharzia species in recreational waters in North-Eastern Poland"

_PeerJ, doi:10.7717/peerj.17598_

## Round 0.1 · original submission · Major Revisions

Dear Dr. Korycińska and colleagues:

Thanks for submitting your manuscript to PeerJ. I have now received three independent reviews of your work, and as you will see, the reviewers raised some concerns about the manuscript. Despite this, these reviewers are optimistic about your work and the potential impact it will have on research studying Trichobilharzia flatworms. Thus, I encourage you to revise your manuscript, accordingly, considering all of the concerns raised by all three reviewers.

While the concerns of the reviewers are relatively minor, this is a major revision to ensure that the original reviewers have a chance to evaluate your responses to their concerns. There are not too many suggestions; thus, it should not take much effort to address these concerns to greatly improve your manuscript.

I look forward to seeing your revision, and thanks again for submitting your work to PeerJ.

Good luck with your revision,

-joe

·

Basic reporting

No comment

Experimental design

Since the infection rate of snalis with animal schistosomes was low in most of the investigated endemic areas, it would be better if more snails were collected and examined in order to obtain more cercariae for examination and comparison. On the other hand, since methods other than shedding, such as the crushing method, have not been used to isolate different stages of schistosoma from snails, it is possible that cases of snail contamination have not been reported.

Validity of the findings

The research method was carried out correctly and carefully, but it is necessary to make corrections in the manuscript, which are mentioned in the additional comments.

Additional comments

Abstract
line 25: Here, it is necessary to mention the methods of isolatingg parasites from the snails.
Line 29: "Two out of 478 (0.4%) L. stagnalis..." Rewrite this paragraph. First, mention that out of 269 snails from the genus Radix, 2 cases of cercariae infection were observed and that both infected snails were identified as Radix auricularia.
In the current form, the reader will assume that the only 2 snails examined in this study were Radix auricularia. While the fact is that you have identified only the infected radixes with the molecular method, not all of them.

Introduction
Line 77: "T. physellae" is correct.
Line81: To write the scientific name in an article, the genus name is mentioned in full only for the first time, the first letter of the genus name is written in the following times. In this case: R. lagotis is correct.
Correct the following scientific names accordingly.
Line 91: Each abbreviation must be defined first. You have not specified in the text what the CD stands for.

Materials and methods
Line 155: Determine from which snails the DNA extraction was done. Of all snails or only infected snails?
Line 182: Change "(Genetics)" to "(Nippon Genetics, Japan)"
Line 186: It would have been better if you had arranged and recorded the sequences in the gene bank at once in one table, so that their accession numbers would be continuous.

Results
Line 212: Change "cercaria" to "cercariae"
Line 216: Change "discovered" to "identified"
Line 221: "was nearly 100% homologous" is not correct. Sequences are either homologous or not (Reeck et al. Cell. 1987;50(5):667).
Instead of "was nearly 100% homologous" you can use "was nearly similar to" or "is most similar to"

Discussion
Lines 255 to 261: This paragraph has nothing to do with the research results and should be deleted.
Line 292: It is necessary to discuss more about the life cycle of T. szidati, T. franki and their final hosts in the study area and nearby areas.
Line 302: If there is information about the effect of climate change and changes in the migration of aquatic birds in the desired area, it is better to discuss it here.

Reviewer 2 ·

Basic reporting

No comment.

Experimental design

No comment.

Validity of the findings

No comment.

Additional comments

A very interesting, original research work of great scientific and practical value. The role and importance of an avian schistosomes (larvae/cercariae) in the pathogenesis of human skin diseases is poorly understood, therefore any new data are valuable and useful.

The structure of the article is correct; properly selected research methods were used, with correct analysis of results and logically presented conclusions.

I only have minor, mainly editing comments.

1. Lines, e.g. 39, 57, 58: „larval forms” or „larval stage” ?,
2. Line 40: “Cercarial dermatitis” – please add “(CD)” - “Cercarial dermatitis (CD)”,
3. Lines 82-84: please add a comma to the scientific names of species (between the date and authors),
4. Rather consistently, it would be necessary to provide entire species names (date and author) for other species (hosts).
5. Lines 194, Table 1/host, Table 3, Figure 2/title, 357: “sp.” and “spp.” – should be in normal font,
6a. Lines e.g. 210, 211, Table 3: “Location” refers to a habitat; “locality” refers to a geographical area.
6b. Figure 1: There is "locations" in the title and "locality" under the map.
7. Line 233: „…prevalence of infections…” – maybe “…prevalence of infections (percentage of infected host) …”
8. Line 401: Family – “Anatidae”: should be in normal font,
9. Line 509: “... genusAenigmomphiscolaKruglov” - should be “genus Aenigmomphiscola Kruglov”,
10. Lines 509, 521, 522, 523: genus should be italicized,
11. Line 520: please add the name of the journal – Journal of Parasitology,
12. References; PeerJ uses an alphabetized reference list and full names of journals.
13. tables 1, 2, : please add commas to the citations (column 6),
14. Table 3: “T. f.” or “T. f.” ?, also “T. sz.”, “T. sp.” ?,
15. Table 10: It's better to put units (%) in the column titles.
16. Please check the citations in the text - PeerJ uses a different system/format.

Reviewer 3 ·

Basic reporting

no comment

Experimental design

no comment

Validity of the findings

no comment

Additional comments

In the reviewed article, the authors examine bird flukes of the Trichobilharzia genus, which are the etiological factors associated with human cercarial dermatitis (so-called swimmer's itch). The aim of the study was to conduct a molecular and phylogenetic analysis of Trichobilharzia species occurring in recreational waters of north-eastern Poland.

Firstly, I believe that the choice of this topic is very appropriate, because the number of existing studies on this problem (especially from eastern and northern Europe) is still small, and the information already published is mostly more or less fragmentary.

Secondly, to my knowledge, the methodology used is correct and adequate to the intended research goal. The number of snails tested is high, even from a statistical point of view.

Finally, I also think that the results obtained are interesting and their reliability, despite obtaining a very low prevalence (although generally comparable to the cited data from other countries), is beyond doubt.
Since English is not my native language, I am unable to assess the linguistic quality of the text. For me, the text is understandable, but I suggest that it be additionally reviewed by a native announcer with a biology education.

Generally, I do not have any significant criticisms, so I do not post detailed comments. Overall, I thought the article was well written and competently done. I believe that the article can be published in PeerJ after additional linguistic correction.

---

## Round 0.2 · accepted · Accept

Dear Dr. Korycińska and colleagues:

Thanks for revising your manuscript based on the concerns raised by the reviewers. I now believe that your manuscript is suitable for publication. Congratulations! I look forward to seeing this work in print, and I anticipate it being an important resource for groups studying Trichobilharzia flatworms. Thanks again for choosing PeerJ to publish such important work.

Best,

-joe

·

Basic reporting

No comment

Experimental design

No comment

Validity of the findings

No comment

Additional comments

No comment

Reviewer 2 ·

Basic reporting

The authors have made a significant work to improve the manuscript.

One more editorial comment, but this can be corrected when proofreading. Table 4: It is better to put the unit (%) in the column titles (“percentage match” and “query coverage”) instead of repeating it with each value.

Experimental design

-

Validity of the findings

-

Additional comments

-